# Efficient Computation for Diagonal of Forest Matrix via Variance-Reduced Forest Sampling

## ABSTRACT

The forest matrix, particularly its diagonal elements, has far-reaching implications in network science and machine learning. The state-of-the-art algorithms for the diagonal of forest matrix computation are based on the fast Laplacian solver. However, these algorithms encounter limitations when applied to digraphs due to the incapacity of the Laplacian solver. To overcome the issue, in this paper, we propose three novel sampling-based algorithms: SCF, SCFV, and SCFV+. Our first algorithm SCF leverages a probability interpretation of the diagonal of the forest matrix and utilizes an expansion of Wilson's algorithm to sample spanning converging forests. To reduce the variance in the forest sampling, we develop two novel variance-reduced techniques. The first technique, leading to the proposal of the SCFV algorithm, is inspired by opinion dynamics in graphs and applies matrix-vector iteration to the spanning forest sampling. While SCFV achieves reduced variance compared to SCF, the cross-product term in its variance expression can be complex and potentially large in certain graphs. Therefore, we develop another technique, leading to a new iteration equation and the SCFV+ algorithm. SCFV+ achieves further reduced variance without the cross-product term in the variance of SCFV. We prove that SCFV+ can achieve a relative error guarantee with high probability and maintain a linear time complexity relative to the nodes of graphs, presenting a superior theoretical result compared to state-of-the-art algorithms. Finally, we conduct extensive experiments on various real-world networks, showing that our algorithms achieve better estimation accuracy and are more time-efficient than the state-of-the-art algorithms. Moreover, our algorithms are scalable to massive graphs with more than twenty million nodes in both undirected and directed graphs.

## KEYWORDS

Forest matrix, Wilson's algorithm, spanning converging forest, variance reduction

## 1 INTRODUCTION

As a typical representation of a graph, the Laplacian matrix $L$ encapsulates much useful structural and dynamical information of the graph [34]. In addition to $L$ itself, the forest matrix, denoted as $\Omega = (I + L)^{-1}$, is also a powerful tool in network science, with close ties to spanning rooted forests in graphs [11, 12]. Recent studies have spotlighted various applications of the forest matrix and its variants, such as in Markov processes [3, 4], opinion dynamics [19, 44, 48], and graph signal processing [36, 37]. Particularly, the diagonal entries of $\Omega$ appear frequently in diverse applications. First, it can serve as the forest closeness centrality [22, 45] of a network. Besides, the diagonal of the forest matrix has been closely associated with determinantal point processes in machine learning [25], and has found relevance through electrical interpretations in multi-agent and network-based problems [40].

In order to achieve better effects of the applications for the diagonal entries of $\Omega$ for a graph with $n$ nodes, the first step is to compute or evaluate the diagonal of $\Omega$. A straightforward computation of $\Omega$ involves inverting matrix $I + L$, which costs $O(n^3)$ operations and $O(n^2)$ memories and thus is prohibitive for relatively large graphs. In previous work, two Laplacian solver [15] based algorithms, JLT and UST, were proposed to compute the diagonal of $\Omega$ [22, 45]. Although these methods outperform the standard approach, they are constrained by the Laplacian solver's inability to handle directed graphs. Moreover, while UST exhibits superior performance compared to JLT in experiments [45], it provides only an absolute error guarantee theoretical analysis, whereas JLT offers a relative error guarantee [22]. Consequently, a theoretically guaranteed estimation algorithm for approximating diagonal of $\Omega$ for both undirected and directed graphs is imperative.

In this paper, we delve deep into the problem of efficiently computing the diagonal of the forest matrix in a digraph $\mathcal{G}(V, E)$ with $n$ nodes, in order to overcome the challenges and limitations of existing algorithms. The main contributions of this work are summarized as follows:

- We introduce two forest interpretations of the diagonal of the forest matrix, from the perspectives of average tree size and rooted probability, respectively. From the probability interpretation, we develop SCF, an algorithm that leverages an expansion of Wilson's algorithm to approximate the diagonal of the forest matrix, offering a time complexity of $O(ln)$, where $l$ represents the sampling number.
- To reduce the variance in forest sampling, we develop two novel variance-reduced techniques and propose SCFV and SCFV+. We prove that the variance of the estimators in the three algorithms diminishes progressively. Notably, SCFV+ not only achieves a relative error guarantee with high probability but also maintains a linear time complexity relative to the nodes of graphs, thereby presenting a superior theoretical result compared to existing algorithms.
- Through extensive experiments on various real-world networks, both undirected and directed, our algorithms achieve better estimation accuracy and enhance time efficiency compared to the state-of-the-art algorithms. Moreover, our algorithms are scalable to massive graphs with over thirty million nodes.

## 2 RELATED WORK

Identifying crucial nodes in a graph is a fundamental issue with a rich history in machine learning and graph analysis [20, 35, 42, 43]. Various metrics and indices have been developed to quantify the relative importance or centrality of nodes within a network [24], including but not limited to, betweenness centrality [17], closeness centrality [5, 6], eccentricity [9]. In addition to the classic centrality measures, forest closeness centrality has been proposed

and explored for its unique advantages [22]. One of the notable advantages of forest closeness centrality is its applicability to disconnected networks, which is particularly relevant for various real-world networks such as Mobile Ad hoc Networks [16] and protein-protein interaction networks [21]. Furthermore, the forest centrality has a better-discriminating power than alternate metrics such as betweenness, harmonic centrality, eigenvector centrality, and PageRank[22].

The calculation of forest closeness centrality is inherently tied to the diagonal elements of the forest matrix $\Omega$. To speed up the computation, a nearly linear time algorithm JLT was proposed in [22], which combines the Johnson-Lindenstrauss lemma [1, 23] with the fast Laplacian solver [15], necessitating a time complexity of $O(m\epsilon^{-2}\log^{2.5} n \log \frac{1}{\epsilon} \text{polyloglog}(n))$ to achieve a relative error bound. Subsequently, the UST algorithm was proposed in [45], combining a single instance of the Laplacian solver and uniform spanning tree sampling. Compared to JLT, UST achieves computational acceleration in experiments and needs a total time complexity of $\widetilde{O}(m\epsilon^{-2}\log^{3/2} n)$ to guarantee an absolute error of $\epsilon$ with high probability. However, both algorithms utilize the fast Laplacian solver, which is not applicable to digraphs.

Wilson's algorithm plays a pivotal role in our three algorithms SCF, SCFV, and SCFV+. Initially proposed to sample a spanning tree in graphs [46, 47], Wilson's algorithm and its variants have found applications across various domains, such as computing the PageRank vector [30, 32], solving linear systems in graph signal processing [36, 37], addressing optimization problems in opinion dynamics [44], and estimating effective resistance [31, 45]. While several variance reduction techniques have been proposed and utilized in various sampling-based problems [30, 38, 39], these techniques are either unsuitable for our diagonal estimation problem or induce a prohibitively high complexity. Thus, developing novel estimators with reduced variance for the diagonal of the forest matrix, applicable to both undirected graphs and digraphs, becomes the primary research subject of this paper.

## 3 PRELIMINARIES

### 3.1 Graph and Laplacian Matrix

Consider an unweighted simple directed graph (digraph) $\mathcal{G} = (V, E)$ with $n = |V|$ nodes (vertices) and $m = |E|$ directed edges (arcs), where $V = \{v_1, v_2, \ldots, v_n\}$ represents the node set, and $E$ signifies the set of directed edges such that $E = \{(v_i, v_j) \mid v_i, v_j \in V\}$. An arc $(v_i, v_j) \in E$ indicates a directed edge pointing from node $v_i$ to node $v_j$. In what follows, $v_i$ and $i$ are used interchangeably to represent node $v_i$ if incurring no confusion. Let $N(i)$ be the node set accessible from node $i$. In other words, $N(i) = \{j : (i, j) \in E\}$. A digraph is called weakly connected if it is connected when one replaces any directed edge $(i, j)$ with two directed edges $(i, j)$ and $(j, i)$ in opposite directions.

The structure information of digraph $\mathcal{G} = (V, E)$ is characterized by its adjacency matrix $\boldsymbol{A} = (a_{ij})_{n\times n}$, where $a_{ij} = 1$ if $(v_i, v_j) \in E$ and $a_{ij} = 0$ otherwise. For any node $i$ in $\mathcal{G}$, its in-degree $d_i^+$ and out-degree $d_i^-$ are given by $d_i^+ = \sum_{j=1}^{n} a_{ji}$ and $d_i^- = \sum_{j=1}^{n} a_{ij}$, respectively. In the sequel, we use $d_i$ to represent the out-degree $d_i^-$. The diagonal out-degree matrix of digraph $\mathcal{G}$ is defined as $\boldsymbol{D} = \text{diag}(d_1, d_2, \ldots, d_n)$, and the Laplacian matrix of digraph $\mathcal{G}$ is defined to be $\boldsymbol{L} = \boldsymbol{D} - \boldsymbol{A}$. Let $\boldsymbol{I}$ be the $n$-dimensional identity matrix, and $\boldsymbol{e}_i$ be the $i$-th standard basis column vector, with $i$-th element being 1 and other elements being 0.

### 3.2 Spanning Converging Forests and Forest Matrix

For a digraph $\mathcal{G} = (V, E)$, a spanning subgraph of $\mathcal{G}$ retains all nodes from $V$ but may only include a subset of edges from $E$. A rooted converging tree is a weakly connected digraph, where one node, called the root node, has an out-degree of 0, and all other nodes have an out-degree of 1. An isolated node is considered as a converging tree with the root being itself. A spanning converging forest of digraph $\mathcal{G}$ that includes all nodes and whose weakly connected components are rooted converging trees. Such a forest aligns with the concept of an in-forest as described in [2, 10].

The forest matrix [12, 13] is defined as $\Omega = (\boldsymbol{I} + \boldsymbol{L})^{-1} = (\omega_{ij})_{n\times n}$. In the context of digraphs, the forest matrix $\Omega$ is row stochastic, with all its components in the interval $[0, 1]$. Moreover, for each row, the diagonal elements surpass the other elements, that is $0 \le \omega_{ji} < \omega_{ii} \le 1$ for any pair of nodes $i$ and $j$, and the diagonal element $\omega_{ii}$ of matrix $\Omega$ satisfies $\frac{1}{1+d_i} \le \omega_{ii} \le \frac{2}{2+d_i}$ [44]. Subsequently, we use $\boldsymbol{\omega}$ to denote the column vector of all diagonal elements of the forest matrix, that is $\boldsymbol{\omega} = (\omega_{11}, \cdots, \omega_{nn})^\top$.

## 4 FAST FOREST SAMPLING ALGORITHM

In this section, we introduce two interpretations of the forest matrix diagonal. Utilizing the probability interpretation, along with an extension of Wilson's algorithm, we propose a fast sampling algorithm to calculate the diagonal elements vector $\boldsymbol{\omega}$ of the forest matrix.

### 4.1 Novel Forest Interpretation for Diagonal of Forest Matrix

In this subsection, we introduce a novel interpretation for the diagonal of forest matrix $\Omega$. Before proceeding, we introduce some essential notations.

For an unweighted digraph $\mathcal{G} = (V, E)$, let $\mathcal{F}$ denote the set of all spanning converging forests. For a given spanning converging forest $\phi \in \mathcal{F}$, define the root set $\mathcal{R}(\phi)$ of $\phi$ as the collection of roots from all converging trees that constitute $\phi$, that is, $\mathcal{R}(\phi) = \{i : (i, j) \notin \phi, \forall j \in V_\phi\}$. Since each node $i$ in $\phi$ is part of a specific converging tree, we define a function $r_\phi(i) : V \to \mathcal{R}(\phi)$ mapping node $i$ to the root of its associated converging tree. Thus, if $r_\phi(i) = j$, it implies that $j$ is in $\mathcal{R}(\phi)$, and both nodes $i$ and $j$ are part of the same converging tree in $\phi$. Define $\mathcal{F}_{ij}$ as the set of spanning converging forests in which nodes $i$ and $j$ are within the same converging tree, rooted at node $j$. Formally, $\mathcal{F}_{ij} = \{\phi : r_\phi(i) = j, \phi \in \mathcal{F}\}$. It follows that $\mathcal{F}_{ii} = \{\phi : i \in \mathcal{R}(\phi), \phi \in \mathcal{F}\}$. For example, the left side of Figure 1 is a toy digraph consisting of 5 nodes and 8 edges, while the right side illustrates one of its spanning converging forests with roots marked in red. Let $\phi$ represent the spanning converging forest shown in Figure 1. With the above notations, we have $\mathcal{R}(\phi) = \{3, 5\}$, and $r_\phi(1) = 3$.

For a node $i \in V$ and a spanning converging forest $\phi \in \mathcal{F}_{ii}$, let $N(\phi, i)$ be a set defined by $N(\phi, i) = \{j : r_\phi(j) = i\}$. By definition,

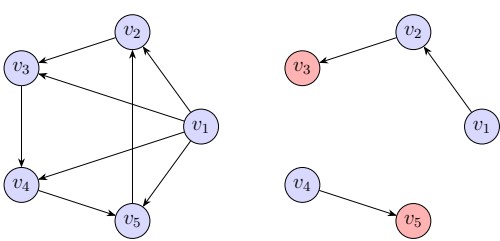

**Figure 1: A toy digraph and one of its spanning converging forest $\phi$.**

for any $\phi \in \mathcal{F}_{ii}$, $|N(\phi, i)|$ is equal to the number of nodes in the converging tree in $\phi$, whose root is node $i$. For two nodes $i$ and $j$ and a spanning converging forest $\phi$, define $\mathbb{I}_{\{r_\phi(i)=j\}}$ as an indicator function, which is 1 if the input statement is true and 0 otherwise. For example, if $r_\phi(i) = j$, $\mathbb{I}_{\{r_\phi(i)=j\}} = 1$, and $\mathbb{I}_{\{r_\phi(i)=j\}} = 0$ otherwise.

With the previously defined notations, we present an interpretation of the diagonal of the forest matrix $\Omega$ in undirected graphs. The following theorem establishes the relationship between the reciprocal of $\omega_{ii}$ and the average number of nodes rooted at node $i$.

THEOREM 4.1. *For an undirected graph $G = (V, E)$, the reciprocal of the $i-th$ diagonal elements of forest matrix $\Omega$ is equal to the average size of the tree containing node $i$ across all converging spanning forest where $i$ serves as one of the root nodes. Formally, this can be expressed as*

$$\frac{1}{\omega_{ii}} = \frac{\sum_{\phi \in \mathcal{F}_{ii}} N(\phi, i)}{|\mathcal{F}_{ii}|}.$$

In [22], the $i$-th diagonal element $\omega_{ii}$ of the forest matrix $\Omega$ is interpreted as the forest distance from node $i$ to other nodes, with a smaller sum of distances indicating a more pivotal node. In Theorem 4.1, we introduce a novel interpretation for $\omega_{ii}$, suggesting that if the average tree size rooted at $i$ is larger, then $\frac{1}{\omega_{ii}}$ will be larger, $\omega_{ii}$ will be smaller, and consequently, node $i$ will be more significant. This aligns with the analysis in [22]. However, both the distance interpretation and the average tree size interpretation are valid only for undirected graphs, as their derivations utilize the symmetry of the forest matrix, a property exclusive to undirected graphs. Subsequently, we propose another interpretation from a probabilistic perspective, which is applicable to both undirected graphs and digraphs and inspires us for the design of our sampling algorithms.

## 4.2 Probability Interpretation and Expansion of Wilson's Algorithm

The entries of the forest matrix are closely related to the spanning converging forest in graphs. Using the approach in [8, 11, 12], the entry $\omega_{ij}$ of the forest matrix $\Omega$ can be expressed as $\omega_{ij} = |\mathcal{F}_{ij}|/|\mathcal{F}|$. By setting $i = j$, for a node $i \in V$, the equation $\omega_{ii} = |\mathcal{F}_{ii}|/|\mathcal{F}|$ holds true. This suggests that a probabilistic interpretation of the diagonal of the forest matrix can be provided, representing the probability of node $i$ being included in the root set $\mathcal{R}(\phi)$ when a spanning converging forest $\phi \in \mathcal{F}$ is sampled uniformly. Then for a spanning converging forest $\phi \in \mathcal{F}$, we can define an estimator

$\widehat{\omega}_{ii}(\phi)$ of $\omega_{ii}$ as $\widehat{\omega}_{ii}(\phi) = \mathbb{I}_{\{i \in \mathcal{R}(\phi)\}}$. The estimator $\widehat{\omega}_{ii}$ is unbiased since

$$\mathbb{E}\{\widehat{\omega}_{ii}(\phi)\} = \mathbb{P}\{i \in \mathcal{R}(\phi)\} = \frac{|\mathcal{F}_{ii}|}{|\mathcal{F}|} = \omega_{ii}.$$

Therefore, if we can uniformly generate a spanning converging forest in $\mathcal{G}$, and record the probability of $i$ serving as a root node, we can estimate $\omega_{ii}$. In the following, we give a brief introduction of an expansion of Wilson's Algorithm in order to uniformly sample spanning converging forest $\phi \in \mathcal{F}$.

Wilson proposed an algorithm based on a loop-erased random walk to get a spanning tree rooted at a given node [46]. The loop-erasure technique, pivotal to this algorithm, is a process derived from the random walk by performing an erasure operation on its loops in chronological order [27, 28]. For a digraph $\mathcal{G} = (V, E)$, we can also apply an expansion of Wilson's Algorithm to get a spanning converging forest $\phi \in \mathcal{F}$, by using the method similar to that in [4, 37, 44], which includes the following steps. Firstly, construct an augmented digraph $\mathcal{G}'$ by adding one new node $x$. Then for each node $i$ in the original graph, add new edge $(i, x)$ to the augmented graph $\mathcal{G}'$. Subsequently, use Wilson's algorithm to generate a rooted spanning tree in the augmented graph $\mathcal{G}'$, designating $x$ as the root node. Finally, delete node $x$ and all edges connected to it in the rooted spanning tree, and define the root set $\mathcal{R}$ as the nodes with an out-degree of 0, thereby obtaining a spanning converging forest in $\mathcal{G}$. For example, in Figure 2, the left side is the augmented toy graph, with one new node $x$ and some new edges added. The middle of Figure 2 illustrates a rooted spanning tree in the augmented graph, rooted at $x$. The right side is a spanning converging forest in the original toy graph, obtained by deleting node $x$ and its connected edges.

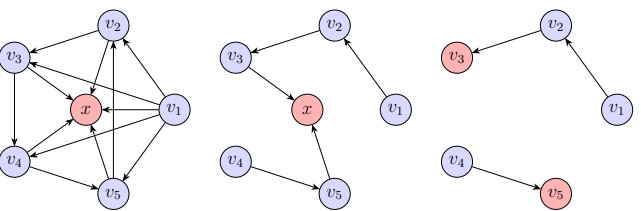

**Figure 2: A spanning converging forest generated using the expansion of Wilson's algorithm for the toy graph.**

Since Wilson's algorithm returns a uniform rooted spanning tree [46], the spanning converging forest obtained using the above steps is also uniformly selected from $\mathcal{F}$.

## 4.3 Fast Sampling Algorithm

In this subsection, we propose a fast sampling algorithm to estimate the diagonal of the forest matrix based on the expansion of Wilson's algorithm mentioned earlier. Additionally, we provide a theoretical analysis concerning its time complexity and relative error guarantee.

As shown above, we have defined an unbiased estimator $\widehat{\omega}_{ii}(\phi)$, and introduced how to employ the expansion of Wilson's algorithm to uniformly generate a spanning converging forest $\phi \in \mathcal{G}$. Then we can generate $l$ spanning forest $\phi_1, \cdots, \phi_l$, and use the average value $\frac{1}{l} \sum_{j=1}^{l} \widehat{\omega}_{ii}(\phi_j)$ to approximate $\omega_{ii}$. We detailed this in Algorithm 1.

**Algorithm 1:** SCF($\mathcal{G}, l$)

**Input** : $\mathcal{G}$:a digraph

$l$:number of generated spanning converging forest

**Output** : $\widehat{\boldsymbol{\omega}}$ : a vector approximating the diagonal elements of the forest matrix

1 **Initialize** : $\widehat{\boldsymbol{\omega}}[i] \leftarrow 0, i = 1, 2, \ldots, n$

2 **for** $t = 1$ to $l$ **do**

3      $\phi_t \leftarrow$ a spanning converging forest generated from $\mathcal{G}$

4      **for** $i = 1$ to $n$ **do**

5          $j \leftarrow r_{\phi_t}(i)$

6          **if** $i = j$ **then**

7              $\widehat{\boldsymbol{\omega}}[i] \leftarrow \widehat{\boldsymbol{\omega}}[i] + \frac{1}{l}$

8 **return** $\widehat{\boldsymbol{\omega}}$

We now analyze the time complexity of Algorithm 1 in the following lemma.

LEMMA 4.2. *For any unweighted digraph $\mathcal{G} = (V, E)$, the expected time complexity of Algorithm 1 is $O(ln)$.*

Lemma 4.2 shows that the time complexity of algorithm 1 is closely related to the sampling number $l$. The estimation of $l$ needs the Chernoff bound [14] and is given in Theorem 4.3.

THEOREM 4.3. *For any node $i$ with $\omega_{ii} > \sigma$, and any parameters $\epsilon, \sigma, \delta \in (0, 1)$, if $l$ is chosen obeying $l = \left\lceil \frac{6+2\epsilon}{3\sigma\epsilon^2} \ln \frac{2}{\delta} \right\rceil$, then the approximation $\widehat{\boldsymbol{\omega}}[i]$ of $\omega_{ii}$ returned by Algorithm 1 satisfies the following relation with the probability of at least $1 - \delta$:*

$$(1 - \epsilon)\omega_{ii} \le \widehat{\boldsymbol{\omega}}[i] \le (1 + \epsilon)\omega_{ii}. \tag{1}$$

From Theorem 4.3, it is evident that for a directed graph $G = (V, E)$, if one desires an $\epsilon$ relative error guarantee using Algorithm 1 with a probability of at least $1 - 1/n$, then the complexity of the algorithm is $O(\frac{n \log n}{\sigma \epsilon^2})$. According to Theorem 4.3, this guarantee is applicable only to those nodes where $\omega_{ii} > \sigma$. However, the situation becomes more complex for graphs containing nodes with high degrees. As shown in [44], $\omega_{ii}$ is upper-bounded by $\frac{2}{2+d_i}$. Consequently, for nodes with significantly large degrees, the number of samples required by Algorithm 1 to achieve the desired relative error guarantee can become prohibitively large.

# 5 A NEW ESTIMATOR WITH REDUCED VARIANCE

In this section, we introduce a novel estimator for $\omega_{ii}$, designed to overcome the challenge encountered in Algorithm 1, where the number of samplings may become excessively large.

## 5.1 Inspiration from FJ Model on Digraphs

The inspiration for the new estimator is drawn from the widely recognized Friedkin-Johnsen (FJ) model [18], a prevalent model for opinion evolution and formation. For the FJ opinion model on a digraph $\mathcal{G} = (V, E)$, each node/agent $i \in V$ is associated with two opinions: one is the internal opinion $s_i$, the other is the expressed opinion $z_i(t)$ at time $t$. The internal opinion $s_i$, which lies within the closed interval $[0, 1]$, represents node $i$'s inherent stance on a

specific topic. During the process of opinion evolution, the internal opinion $s_i$ remains constant, while the expressed opinion $z_i(t)$ evolves at time $t + 1$ as follows:

$$z_i(t + 1) = \frac{s_i + \sum_{j \in N(i)} z_j(t)}{1 + d_i}. \tag{2}$$

Let $\boldsymbol{s} = (s_1, s_2, \cdots, s_n)^\top$ denote the vector of internal opinions, and let $\boldsymbol{z}(t) = (z_1(t), z_2(t), \cdots, z_n(t))^\top$ denote the vector of expressed opinions at time $t$. The following lemma reveals the convergence result of the iteration.

LEMMA 5.1. *[7] Regardless of the initial value of $\boldsymbol{z}(0)$, if $\boldsymbol{z}(t + 1)$ evolves according to the equation in (2), where $t = 1, 2, \cdots$, then as $t$ approaches infinity, $\boldsymbol{z}(t)$ converges to an equilibrium vector $\boldsymbol{z} = (z_1, z_2, \cdots, z_n)^\top$ satisfying $\boldsymbol{z} = \Omega \boldsymbol{s}$.*

Lemma 5.1 elucidates the correlation between the equilibrium expressed opinion $\boldsymbol{z} = (z_1, z_2, \cdots, z_n)^\top$ and the initial opinion $\boldsymbol{s}$, with the forest matrix $\Omega$ playing a pivotal role in this relationship. Consequently, Lemma 5.1 offers a novel approach to determine the diagonal elements of the forest matrix. Let $\boldsymbol{\rho}^i = \Omega \boldsymbol{e}_i$ represent the $i$-th column of $\Omega$. Then the $i$-th diagonal element of $\Omega$ is exactly the $i$-th element $\rho_i^i$ of vector $\boldsymbol{\rho}^i$. To obtain the diagonal element $\omega_{ii}$, we can initially set the opinion vector $\boldsymbol{s} = \boldsymbol{e}_i$ and select an appropriate vector $\boldsymbol{\rho}^i(0)$. Then repeat the iteration equation in (2) $t$ times to yield $\boldsymbol{\rho}^i(t)$. According to Lemma 5.1, the $i$-th component $\rho_i^i(t)$ of $\boldsymbol{\rho}^i(t)$ serves as an estimator for $\omega_{ii}$, and as $t$ increases, the discrepancy between $\rho_i^i(t)$ and $\omega_{ii}$ diminishes.

However, employing the iteration procedure to compute all diagonal elements of the forest matrix presents several challenges. For a fixed node $i$, executing the iteration equation $t$ times needs a time complexity of $O(mt)$. Given that all $n$ diagonal elements require computation, the naive iteration approach demands a time complexity of $O(mnt)$, which is computationally exhaustive.

## 5.2 A Novel Unbiased Estimator

To address this challenge, it is pertinent to note that for a specific node $i$, the required number of iterations $t$ to achieve an error bound between $\boldsymbol{\rho}^i$ and $\boldsymbol{\rho}^i(t)$ varies significantly with the initial vector $\boldsymbol{\rho}^i(0)$. Specifically, if $\boldsymbol{\rho}^i(0)$ precisely matches the equilibrium vector $\boldsymbol{\rho}^i = \Omega \boldsymbol{e}_i$, the iterative equation will maintain the value of $\boldsymbol{\rho}^i$ unchanged for any $t = 1, \cdots$, as $\boldsymbol{\rho}^i$ is the system's equilibrium vector. In this scenario, the required iteration number $t$ is zero. However, the exact value of $\boldsymbol{\rho}^i$ is typically unknown. Adopting a similar idea, we can initialize $\boldsymbol{\rho}^i(0)$ as an easily obtainable estimator of $\boldsymbol{\rho}^i$, which brings $\boldsymbol{\rho}^i$ and $\boldsymbol{\rho}^i(0)$ closer initially, thereby reducing the number of iterations $t$ needed.

Wilson's algorithm plays an important role again. For a spanning converging forest $\phi \in \mathcal{F}$, define a random variable $\widehat{\omega}_{ji}$ as $\widehat{\omega}_{ji}(\phi) \triangleq \mathbb{I}_{\{r_\phi(j)=i\}}$. The estimator $\widehat{\omega}_{ji}$ is an unbiased estimator of $\omega_{ji}$ if we randomly select $\phi$ since $\mathbb{E}\{\widehat{\omega}_{ji}(\phi)\} = \mathbb{P}\{r_\phi(j) = i\} = |\mathcal{F}_{ji}|/|\mathcal{F}| = \omega_{ij}$. Then we can generate $l$ spanning converging forests $\phi_1, \cdots, \phi_l$, and set the initial iteration vector $\boldsymbol{\rho}^i(0) = (\rho_1^i(0), \cdots, \rho_n^i(0))$, where $\rho_j^i(0) = \frac{1}{l} \sum_{k=1}^{l} \widehat{\omega}_{ji}(\phi_k)$. Then, we repeat the iteration $t$ times and get $\boldsymbol{\rho}^i(t)$, which serves as an estimator for $\boldsymbol{\rho}^i$.

Recall that our objective is to reduce the iteration times, focusing on the $i$-th component of $\boldsymbol{\rho}^i$. A bold and natural idea emerges:

What if we only perform one iteration? This case offers a unique perspective that might be the key to our challenge. In this scenario, the estimator $\rho^i(t)$ can be elegantly expressed as: $\rho^i(1) = (I + D)^{-1}e_i + (I + D)^{-1}A\rho^i(0)$. Moreover, since no further iterations are needed, we only need to focus on the $i$-th component of $\rho^i(1)$, and do not need to calculate the other elements. That is, we only need to do calculate that

$$\rho_i^i(1) = \frac{1}{1+d_i}\big(1 + \sum_{j \in N(i)} \rho_j^i(0)\big) = \frac{1}{1+d_i}\big(1 + \frac{1}{l}\sum_{j \in N(i)}\sum_{k=1}^{l} \widehat{\omega}_{ji}(\phi_k)\big).$$

From the expression of $\rho_i^i(1)$, we can define a new estimator as

$$\widetilde{\omega}_{ii}(\phi) \triangleq \frac{1}{1+d_i}\big(1 + \sum_{j \in N(i)} \widehat{\omega}_{ji}(\phi)\big).$$

Then we derive that $\rho_i^i(1) = \frac{1}{l}\sum_{k=1}^{l}\widetilde{\omega}_{ii}(\phi_k)$. That is, we can directly use Wilson's algorithm to sample $l$ spanning converging forests, and then obtain the value of $\widetilde{\omega}_{ii}(\phi_k)$. The average of $l$ values is equal to the $i$-th component of $\rho^i(1)$. We detail this in Algorithm 2, demonstrating the method to derive this novel variable. The algorithm takes a parameter $l$, which signifies the number of spanning converging forests to be sampled. It then returns a vector $\widetilde{\omega}$ as an estimator for the diagonal elements of the forest matrix.

---

**Algorithm 2:** SCFV($\mathcal{G}, l$)

**Input** : $\mathcal{G}$:a digraph

$\quad\quad\quad$ $l$:number of generated spanning converging forest

**Output** : $\widetilde{\omega}$ : a vector approximating the diagonal elements of the forest matrix

1 **Initialize** : $\widetilde{\omega}[i] \leftarrow \frac{1}{1+d_i}$, $i = 1, 2, \ldots, n$

2 **for** $t = 1$ *to* $l$ **do**

3 $\quad$ $\phi_t \leftarrow$ a spanning converging forest generated from $\mathcal{G}$

4 $\quad$ **for** $i = 1$ *to* $n$ **do**

5 $\quad\quad$ $j \leftarrow r_{\phi_t}(i)$

6 $\quad\quad$ **if** $i \in N(j)$ **then**

7 $\quad\quad\quad$ $\widetilde{\omega}[j] \leftarrow \widetilde{\omega}[j] + \frac{1}{l(1+d_j)}$

8 **return** $\widetilde{\omega}$

---

The algorithm starts by initializing $\widetilde{\omega}_{ii}(\phi)$ to 0. It then aims to generate $l$ spanning converging forests using Wilson's Algorithm. After generating each forest $\phi_t$, a loop is executed to update the vector $\widetilde{\omega}$. Similar to the analysis of Lemma 4.2, the total computational complexity of Algorithm 2 is $O(ln)$. Then, we propose a lemma, which shows that $\widetilde{\omega}_{ii}(\phi_k)$ is still an unbiased estimator while having less variance.

LEMMA 5.2. *For node* $i \in V$, $\widetilde{\omega}_{ii}(\phi)$ *is an unbiased estimator of* $\omega_{ii}$. *Let* $q_{jk}^{(i)}$ *represent the ratio of spanning converging forests where both roots of $j$ and $k$ are node $i$ to the total number of such forests. The variance of this estimator is given by*

$$\text{Var}\{\widetilde{\omega}_{ii}(\phi)\} = \frac{3\omega_{ii}}{1+d_i} - \frac{2}{(1+d_i)^2} + \frac{2\sum_{j,k \in N(i)} q_{jk}^{(i)}}{(1+d_i)^2} - \omega_{ii}^2.$$

*Importantly, this variance is always less than or equal to the variance of the estimator* $\widehat{\omega}_{ii}$.

Lemma 5.2 highlights the reduced variance of the random variable $\widetilde{\omega}_{ii}(\phi)$. However, a challenge arises when we invoke the Chernoff bound to determine the requisite sampling number. The third term of the variance, namely $\frac{2\sum_{j,k \in N(i)} q_{jk}^{(i)}}{(1+d_i)^2}$, is inherently complex. Deriving a proper upper bound for this term is not straightforward, which complicates the task of determining an appropriate sampling number $l$. Consequently, although the new estimator $\widetilde{\omega}_{ii}(\phi)$ facilitates a reduction in variance, we are unable to obtain a satisfying theoretical result for the sampling number $l$ due to the complexities introduced by this term.

## 6 NEW ITERATION EQUATION FOR FURTHER VARIANCE REDUCTION

In this section, we introduce a new iteration equation and propose a superior estimator. This novel estimator notably omits the complex cross-product term found in the variance of $\widetilde{\omega}_{ii}(\phi)$, in order to further reduce the variance in samplings, and derive a better theoretical result.

### 6.1 A New Iteration Equation

To further refine our estimator, it is insightful to revisit the estimator $\widetilde{\omega}_{ii}(\phi)$, which draws inspiration from Equation (2). In (2), the update of each node is influenced by its neighbors, specifically, $z_i(t+1)$ updates according to the value of $z_j(t)$ where $j \in N(i)$. A novel idea emerges when we consider inverting the direction of opinion dissemination, implying that $z_i(t+1)$ updates according to the value of $z_j(t)$ where $i \in N(j)$. In this scenario, our focus shifts to the $i$-th row of the forest matrix $\Omega$, denoted by $\gamma^{i\top} \triangleq e_i^\top\Omega$. The $i$-th component of $\gamma^i$ precisely corresponds to the $i$-th diagonal element of the forest matrix. To compute $\gamma^i$, we propose an iterative equation similar to (2), as follows:

$$\gamma^{i\top}(t+1) = e_i^\top(I+D)^{-1} + \gamma^{i\top}(t)A(I+D)^{-1}. \quad (3)$$

Given that the matrix $(I + L)$ is reversible, the matrix $(I + L^\top)$ is also reversible. Consequently, the iteration in (3) converges to $\gamma^{i\top}$. Consistent with the previous analysis, we aim to restrict the process to a single iteration, as multiple iterations are time unaffordable.

To realize this, we continue to leverage Wilson's algorithm to obtain $\gamma^i(0)$, which serves as a preliminary estimation of the $i$-th row of the forest matrix. Specifically, we can employ Wilson's algorithm to generate $l$ spanning converging forests $\phi_1, \cdots, \phi_l$, and set the initial iteration vector $\gamma^i(0) = (\gamma_1^i(0), \cdots, \gamma_n^i(0))$, where $\gamma_j^i(0) = \frac{1}{l}\sum_{k=1}^{l}\widehat{\omega}_{ij}(\phi_k)$. Upon performing one time of iteration according to (3), we derive that

$$\gamma_i^i(1) = \frac{1}{1+d_i}\big(1 + \sum_{j:i \in N(j)} \gamma_j^i(0)\big) = \frac{1}{1+d_i}\big(1 + \frac{1}{l}\sum_{j:i \in N(j)}\sum_{k=1}^{l} \widehat{\omega}_{ij}(\phi_k)\big).$$

With this new formulation, we can define a novel random variable, $\overline{\omega}_{ii}(\phi)$, for any spanning converging forest $\phi \in \mathcal{F}$ and $i \in V$ as

$$\overline{\omega}_{ii}(\phi) \triangleq \frac{1}{1+d_i} + \frac{1}{1+d_i}\sum_{j:i \in N(j)} \widehat{\omega}_{ij}(\phi).$$

Subsequently, it follows that $\gamma_i^i(1) = \frac{1}{l}\sum_{k=1}^{l}\overline{\omega}_{ii}(\phi_k)$. This implies that we can directly employ Wilson's algorithm to sample $l$

spanning converging forests, and then obtain the value of $\overline{\omega}_{ii}(\phi_k)$. The average of $l$ values is equal to the $i$-th component of $\boldsymbol{\gamma}^i(1)$. We detail this in Algorithm 3, which demonstrates the method to derive this novel variable. The time complexity of Algorithm 3 is $O(ln)$, where $l$ represents the number of spanning converging forests sampled.

---

**Algorithm 3:** SCFV+$(\mathcal{G}, l)$

**Input** : $\mathcal{G}$ : a digraph
$\quad\quad\quad\quad$ $l$:number of generated spanning converging forest
**Output** : $\overline{\omega}$ : a vector approximating the diagonal elements
$\quad\quad\quad\quad\quad$ of the forest matrix
1 **Initialize** : $\overline{\omega}[i] \leftarrow \frac{1}{1+d_i}$, $i = 1, 2, \ldots, n$
2 **for** $t = 1$ *to* $l$ **do**
3 $\quad$ $\phi_t \leftarrow$ a spanning converging forest generated from $\mathcal{G}$
4 $\quad$ **for** $i = 1$ *to* $n$ **do**
5 $\quad\quad$ $j \leftarrow r_{\phi_t}(i)$
6 $\quad\quad$ **if** $i \in N(j)$ **then**
7 $\quad\quad\quad$ $\overline{\omega}[i] \leftarrow \overline{\omega}[i] + \frac{1}{l(1+d_i)}$
8 **return** $\overline{\omega}$

---

## 6.2 Advanced Estimator Analysis and Implementation

We have proposed a novel estimator $\overline{\omega}_{ii}$ for $\omega_{ii}$. Now we introduce a lemma, which shows that this new estimator $\overline{\omega}_{ii}(\phi)$ is superior to $\widetilde{\omega}_{ii}(\phi)$.

LEMMA 6.1. *For any node $i \in V$, $\overline{\omega}_{ii}(\phi)$ serves as an unbiased estimator of $\omega_{ii}$. The variance of this estimator is given by:* $\mathrm{Var}\{\overline{\omega}_{ii}(\phi)\} = \frac{3\omega_{ii}}{1+d_i} - \frac{2}{(1+d_i)^2} - \omega_{ii}^2$. *Crucially, this variance is consistently less than or equal to the variance of the estimator $\widetilde{\omega}_{ii}(\phi)$, and by extension, less than or equal to the variance of the estimator $\widehat{\omega}_{ii}(\phi)$.*

In contrast to $\widetilde{\omega}_{ii}$, the variance of $\overline{\omega}_{ii}$ does not include the complex cross-product term, which previously posed significant challenges when deriving its upper bound. The absence of this complex term in the new estimator simplifies our analysis, as highlighted in the subsequent lemma. This lemma illuminates a significant attribute of the estimator $\overline{\omega}_{ii}$. Specifically, it delineates an upper bound for the ratio of the variance of $\overline{\omega}_{ii}$ to the square of $\omega_{ii}$, a crucial factor when determining the sampling number.

LEMMA 6.2. *Given a directed graph $\mathcal{G} = (V, E)$, for any node $i \in V$, the ratio $\frac{\mathrm{Var}\{\overline{\omega}_{ii}\}}{\omega_{ii}^2}$ is constrained by $\frac{1}{8}$. Formally, $\frac{\mathrm{Var}\{\overline{\omega}_{ii}\}}{\omega_{ii}^2} \leq \frac{1}{8}$.*

Armed with the insights from Lemma 6.2, we proceed to present a theorem that establishes a connection between the relative error bound and the number of samples in Algorithm 3.

THEOREM 6.3. *For any $\epsilon \in (0, 1)$ and $\delta \in (0, 1)$, if $l$ is chosen obeying $l = \left\lceil (\frac{2}{3\epsilon} + \frac{1}{4\epsilon^2}) \log(\frac{2}{\delta}) \right\rceil$, then the approximation $\overline{\omega}[i]$ of $\omega_{ii}$ returned by Algorithm 3 satisfies the following relation with probability of at least $1 - \delta$:*

$$(1 - \epsilon)\omega_{ii} \leq \overline{\omega}[i] \leq (1 + \epsilon)\omega_{ii}. \quad (4)$$

Based on Theorem 3, when we fix the fail probability $\delta$, and the relative error parameter $\epsilon$, the required number of samples remains invariant regardless of the size and structure of the graph. This theoretical insight highlights the superiority of the estimator $\overline{\omega}$ over both $\widehat{\omega}$ and $\widetilde{\omega}$.

## 7 EXPERIMENTS

In this section, we conduct extensive experiments on various real-life networks in order to evaluate the performance of our three algorithms SCF, SCFV, and SCFV+ in terms of effectiveness and efficiency. The source code is publicly available on https://anonymous.4open.science/r/Diagonal-of-Forest-Matrix.

## 7.1 Setup

**Datasets and Equipment.** The datasets of selected real networks are publicly available in the KONECT [26] and SNAP [29]. Our study encompasses a diverse range of networks, both undirected and directed, including but not limited to social networks and road networks. Within these datasets, the number $n$ of nodes ranges from about 16 thousand to 33 million, and the number $m$ of directed edges ranges from about 25 thousand to 301 million. The details of datasets are presented in Table 1. All experiments are conducted using the Julia programming language on a single-threaded setup. We conduct all experiments in a computational environment featuring a 4.2 GHz Intel i7-7700 CPU with 64GB of primary memory.

**Algorithms.** In evaluating the computation of the diagonal elements of the forest matrix, we consider our three proposed algorithms SCF, SCFV, and SCFV+ with two state-of-the-art algorithms, namely JLT [22] and UST [45]. While the two state-of-the-art algorithms, JLT and UST, are confined to undirected graphs due to the limitations of the Laplacian solver they employ, our proposed algorithms SCF, SCFV, and SCFV+ are applicable to both undirected graphs and digraphs.

**Table 1: Datasets used in experiments**

| Type | Dataset | $n$ | $m$ |
|---|---|---|---|
| undirected graphs | web-webbase-2001 | 16,062 | 25,593 |
| | soc-gemsec-RO | 41,773 | 125,826 |
| | tech-p2p-gnutella | 62,561 | 147,878 |
| | tech-RL-caida | 190,914 | 607,610 |
| | soc-twitter-follows | 404,719 | 713,319 |
| | soc-delicious | 536,108 | 1,375,961 |
| | dblp | 5,624,219 | 12,282,055 |
| | livejournal | 7,489,073 | 112,307,315 |
| | delicious | 33,777,767 | 301,183,342 |
| directed graphs | wikipedialinks | 17,649 | 296,918 |
| | p2p-gnutella31 | 62,586 | 147,892 |
| | email-euall | 265,009 | 418,956 |
| | web-Stanford | 281,903 | 2,312,500 |
| | web-Google | 875,713 | 5,105,039 |
| | northwestUSA | 1,207,945 | 2,820,774 |
| | wikitalk | 2,394,385 | 5,021,410 |
| | greatlakes | 2,758,119 | 6,794,808 |
| | fullUSA | 23,947,347 | 57,708,624 |

## 7.2 Experiments on Undirected Graphs

In this subsection, a comparative analysis is conducted between our proposed algorithms SCF, SCFV, and SCFV+ and two state-of-the-art algorithms, namely JLT and UST, focusing on undirected graphs. Initially, experiments are performed on six small to medium-sized networks. For ease of discussion, we employ lowercase letters a through f to denote the six networks: web-webbase-2001(a), soc-gemsec-RO(b), tech-p2p-gnutella(c), tech-RL-caida(d), soc-twitter-follows(e), and soc-delicious(f). For graphs (a), (b), and (c), the ground truth of the diagonal elements of $\Omega$ is computed by directly inverting the matrix $I + L$. Conversely, for graphs (d), (e), and (f), the Conjugate Gradient (CG) solver with a tolerance of $10^{-9}$ is utilized, as direct matrix inversion is computationally infeasible. This approach aligns with the settings in [45].

In evaluating the results, two metrics are considered: average relative errors and maximum relative errors, both assessed over $n$ nodes. Regarding parameter settings, there are primarily two parameters to consider: the sampling number $l$ utilized in UST, SCF, SCFV, and SCFV+, and the dimension $k$ applied in the Johnson-Lindenstrauss (JL) lemma within JLT. Initially, $l$ is set to 500 for the four sampling-based algorithms, and $k$ is set to 50 for JLT. The results of these settings are depicted in Figure 3.

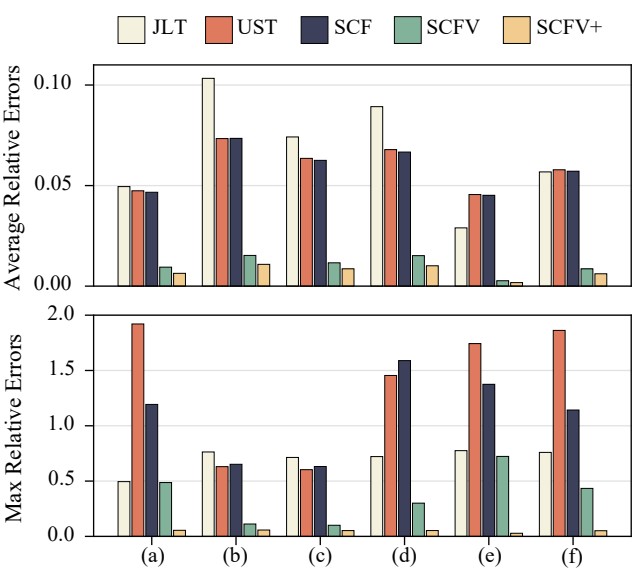

Figure 3: Comparison of maximum (lower) and average (upper) relative errors across five algorithms on six undirected graphs

The upper portion of Figure 3 depicts the average relative errors, while the lower portion illustrates the maximum relative errors across the five algorithms. Observing the average relative errors, it is evident that JLT, UST, and SCF yield similar error results. In contrast, SCFV and SCFV+ achieve nearly 10 times better accuracy with the same number of samples compared to UST and SCF, with SCFV+ securing the best result due to its minimized variance. Considering the maximum relative errors, it is noteworthy that, under these parameter settings, only SCFV+ consistently delivers stable and

satisfactory results across the six graphs. The other algorithms, to varying degrees, yield results that may be deemed inaccurate.

Subsequently, to delve deeper into the effectiveness and efficiency of the five algorithms, we vary the parameters and construct scatter plots. We set $l = 500, 1000, 2000$ and $k = 50, 100, 150$ to observe the resultant effects, which are displayed in Figure 4.

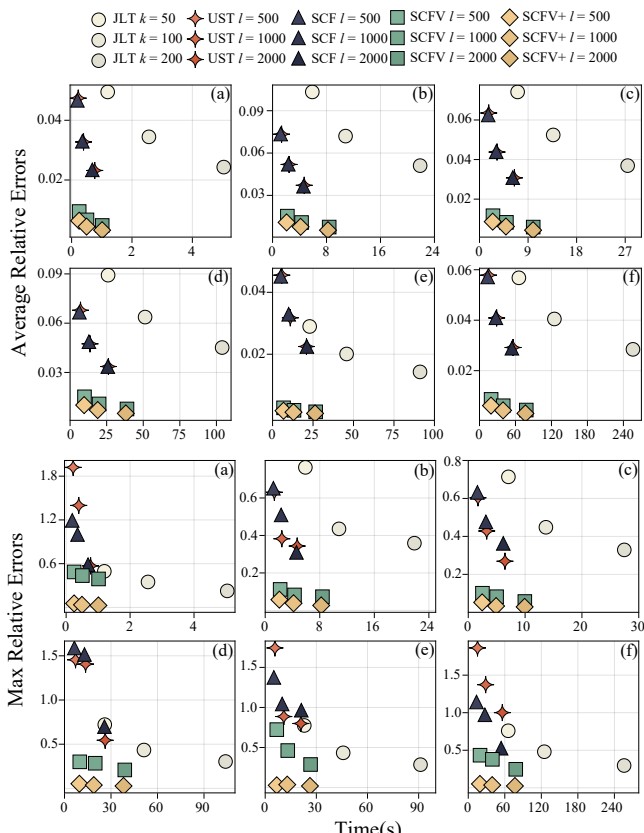

Figure 4: Scatter plot of maximum (lower) and average(upper) relative errors over time for five algorithms on six undirected graphs, considering three parameters each

From Figure 4, it is observable that as the parameters $l$ and $k$ increase, all algorithms demand more time yet yield improved results. Among the five algorithms, JLT exhibits the least optimal performance, as its time requirement escalates rapidly, and its error is not satisfactory, whether considering average relative errors or maximum relative errors.

The remaining four algorithms all rely on sampling using Wilson's algorithm. Observing that if we fix the number of samples, SCF operates slightly faster than UST since UST requires performing one instance of a fast Laplacian solver while SCF does not. Moreover, SCF achieves results comparable to UST. With the same samples, SCFV and SCFV+ require slightly more time than UST and SCF, while the results returned by them are superior.

Among the five algorithms, SCFV+ emerges as the best, as evident from the figures; setting $l = 500$ in SCFV+ can achieve lower errors and faster speeds than the other three sampling algorithms

with $l = 2000$ and JLT with $k = 200$. In conclusion, SCFV+ outperforms the others, whether considering average relative errors or maximum relative errors, and demonstrates stability and satisfactory results.

In the subsequent analysis, we focus on three larger undirected graphs: dblp, livejournal, and delicious. As outlined in Table 1, these graphs are notably substantial, each featuring over 5 million nodes and surpassing 12 million edges. Given time and storage constraints, obtaining the ground truth answer was unfeasible. Similarly, the algorithms JLT and UST fail to run on our equipment due to the time and storage limitations associated with the Laplacian solver, which is utilized in both JLT and UST. Consequently, we perform our three sampling algorithms SCF, SCFV and SCFV+, recording the running time, with the results tabulated in Table 2.

**Table 2: Running time of SCF, SCFV and SCFV+ on three large undirected networks.**

| Graphs | Time(s) | | | | | |
|---|---|---|---|---|---|---|
| | $l = 500$ | | | $l = 1000$ | | |
| | SCF | SCFV | SCFV+ | SCF | SCFV | SCFV+ |
| dblp | 377 | 512 | 518 | 765 | 1034 | 1060 |
| livejournal | 321 | 508 | 541 | 669 | 1030 | 1051 |
| delicious | 1649 | 2437 | 2513 | 3262 | 4872 | 4898 |

An examination of Table 2 indicates that our three algorithms exhibits admirable scalability, performing adeptly on the three expansive networks.

## 7.3 Experiments on Digraphs

In this subsection, experiments are conducted on several directed networks, focusing on the evaluation of the proposed algorithms: SCF, SCFV, and SCFV+. The algorithms JLT and UST are excluded from comparison due to their inapplicability to digraphs. For the determination of the ground truth of the diagonal elements of $\Omega$, the GMRES algorithm [41] is employed with a tolerance set to $10^{-9}$.

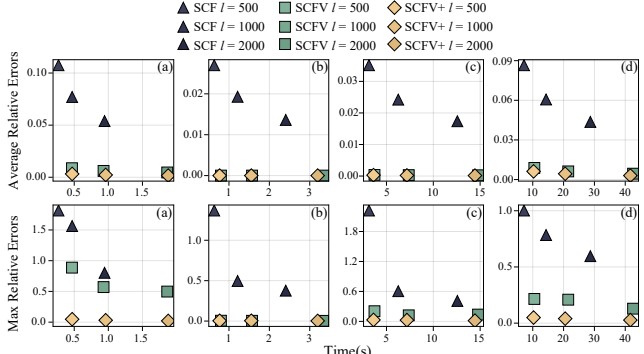

**Figure 5: Scatter plot of maximum (lower) and average (upper) relative errors over time for three algorithms on four digraphs, considering three sampling numbers**

Initially, experiments are executed on four small to medium-sized graphs: wikipedialinks(a), p2p-gnutella31(b), email-euall(c),

and web-Stanford(d). The results of these experiments are depicted in Figure 5.

Figure 5 demonstrates a reduction in error as $l$ increases. Besides, the algorithms SCF, SCFV, and SCFV+ exhibit progressive results, with SCFV+ outperforming the other two in both average and maximum relative errors. While SCFV and SCFV+ yield satisfactory results in terms of average relative errors, only SCFV+ demonstrates credible performance when considering maximum relative errors. This is consistent with the theoretical analysis that the variance of SCFV+ excludes the cross-product term present in the variance of SCFV, as outlined in Lemma 5.2 and Lemma 6.1. Notably, SCFV+ with $l = 500$ surpasses the performance of SCF and SCFV with $l = 2000$, aligning with experimental results obtained from undirected graphs.

Subsequent experiments are conducted on five large digraphs: web-Google, northwestUSA, wikitalk, greatlakes, and fullUSA. Due to limitations in time and storage, the calculation of ground truth is not feasible. Nevertheless, the three proposed algorithms demonstrate commendable scalability and efficacy. Particularly, fullUSA, which comprises more than 23 million nodes and 57 million edges, is processed by our three algorithms within approximately 12 minutes when $l = 500$.

**Table 3: Running time of SCF, SCFV and SCFV+ on five large directed networks.**

| Graphs | Time(s) | | | | | |
|---|---|---|---|---|---|---|
| | $l = 500$ | | | $l = 1000$ | | |
| | SCF | SCFV | SCFV+ | SCF | SCFV | SCFV+ |
| web-Google | 24 | 33 | 32 | 48 | 67 | 66 |
| northwestUSA | 27 | 36 | 34 | 54 | 72 | 70 |
| wikitalk | 18 | 20 | 20 | 35 | 39 | 38 |
| greatlakes | 63 | 84 | 82 | 126 | 168 | 163 |
| fullUSA | 545 | 731 | 710 | 1085 | 1461 | 1417 |

## 8 CONCLUSIONS

In this paper, we addressed the problem of efficiently computing the diagonal of the forest matrix in digraphs. We proposed there novel sampling-based algorithms: SCF, SCFV, and SCFV+. The SCF algorithm utilizes an expansion of Wilson's algorithm, capitalizing on a probabilistic interpretation of the diagonal of the forest matrix. Drawing inspiration from the FJ model, SCFV refines our approach by reducing the variance in forest sampling through the matrix-vector iteration. Our third algorithm SCFV+ further reduces the variance using a novel iteration equation. Notably, SCFV+ achieves a relative error guarantee with high probability and maintains a linear time complexity relative to the nodes of graphs, presenting a superior theoretical result compared to existing algorithms.

We conducted extensive experiments on various real-world networks. Our algorithms demonstrated superior effectiveness and efficiency compared to the state-of-the-art algorithms in undirected graphs. While state-of-the-art algorithms falter in the context of digraphs, our algorithms consistently perform well. Moreover, our algorithms are scalable to massive graphs with over thirty million nodes. In our future work, we plan to extend or improve our algorithm to sign graphs or temporal graphs.

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

## A APPENDIX

In this section we provide proofs of lemmas and theorems in the article.

### A.1 Chernoff Bound

LEMMA A.1. *(Chernoff bound) Let $x_i (1 \leq i \leq l)$ be independent random variables satisfying $|x_i - \mathbb{E}\{x_i\}| \leq M$ for all $1 \leq i \leq l$. Let $x = \frac{1}{l} \sum_{i=1}^{l} x_i$. Then we have*

$$\mathbb{P}\{|x - \mathbb{E}\{x\}| \leq \epsilon\} \geq 1 - 2\exp\left(-\frac{l\epsilon^2}{2(\text{Var}\{x\}l + M\epsilon/3)}\right). \quad (5)$$

### A.2 Proof of Theorem 4.1

**Proof.** Given that $\sum_{j=1}^{n} \omega_{ji} = 1$ and $\omega_{ij} = |\mathcal{F}_{ij}|/|\mathcal{F}|$, the following can be deduced:

$$\frac{1}{\omega_{ii}} = \frac{\sum_{j=1}^{n} \omega_{ji}}{\omega_{ii}} = \frac{1}{|\mathcal{F}_{ii}|} \sum_{j=1}^{n} |\mathcal{F}_{ji}| = \frac{1}{|\mathcal{F}_{ii}|} \sum_{j=1}^{n} \sum_{\phi \in \mathcal{F}_{ii}} \mathbb{I}_{\{r_\phi(j)=i\}}$$

$$= \frac{1}{|\mathcal{F}_{ii}|} \sum_{\phi \in \mathcal{F}_{ii}} \sum_{j=1}^{n} \mathbb{I}_{\{r_\phi(j)=i\}} = \frac{1}{|\mathcal{F}_{ii}|} \sum_{\phi \in \mathcal{F}_{ii}} |N(\phi, i)|, \quad (6)$$

This implies that $\frac{1}{\omega_{ii}}$ represents the average number of nodes within the converging trees that are rooted at node $i$ across all $\phi \in \mathcal{F}_{ii}$. □

### A.3 Proof of Lemma 4.2

**Proof.** Wilson showed that the expected running time of generating a uniform spanning tree of a connected digraph $\mathcal{G}$ rooted at node $u$ is equal to a weighted average of commute times between the root and the other nodes [46]. Marchal rewrote this average of commute times in terms of graph matrices in Proposition 1 in [33]. In [44], the author further analyzed the expected time complexity for the expansion of Wilson's algorithm in $\mathcal{G}'$ is $O(n)$. Thus, the expected time complexity of Algorithm 1 is $O(ln)$. □

### A.4 Proof of Theorem 4.3

**Proof.** Recall that for a given spanning converging forest, denoted as $\phi \in \mathcal{F}$, and a node $i \in V$, the random variable $\widehat{\omega}_{ii}(\phi)$ is defined as $\mathbb{I}_{\{i \in \mathcal{R}(\phi)\}}$. Given that the possible values of $\widehat{\omega}_{ii}(\phi)$ are either 0 or 1, it follows that $|\widehat{\omega}_{ii} - \omega_{ii}| \leq 1$.

In the context of Algorithm 1, we generate a set of $l$ spanning forests, represented as $\phi_1, \cdots, \phi_l$. The output $\widehat{\omega}[i]$ of Algorithm 1 is then defined as: $\widehat{\omega}[i] = \frac{1}{l} \sum_{j=1}^{l} \widehat{\omega}_{ii}(\phi_j)$.

From this, we can compute the variance of $\widehat{\omega}[i]$ as:

$$\text{Var}\{\frac{1}{l} \sum_{j=1}^{l} \widehat{\omega}_{ii}(\phi_j)\} = \frac{1}{l}\text{Var}\{\widehat{\omega}_{ii}(\phi)\} = \frac{1}{l}(\omega_{ii} - \omega_{ii}^2)$$

.

To obtain a relative error bound, it is necessary to satisfy the inequality:

$$\mathbb{P}\{|\widehat{\omega}[i] - \omega_i| \geq \epsilon\omega_i\} \leq \delta.$$

By invoking the Chernoff bound as presented in Lemma A.1, and designating $x_j = \widehat{\omega}_{ii}(\phi_j)$ for $1 \leq j \leq l$ and $x = \widehat{\omega}[i]$, we only need to meet the inequality that:

$$2\exp\left(-\frac{l\epsilon^2\omega_{ii}^2}{2(\text{Var}\{\widehat{\omega}_{ii}\} + M\epsilon\omega_{ii}/3)}\right) \leq \delta,$$

which leads to:

$$l \geq \log(\frac{2}{\delta})\left(\frac{2\text{Var}\{\widehat{\omega}_{ii}\}}{\epsilon^2\omega_{ii}^2} + \frac{2M}{3\epsilon\omega_{ii}}\right). \quad (7)$$

Since $|\widehat{\omega}_{ii} - \omega_{ii}| \leq 1$, we can set $M = 1$. Considering that $\text{Var}\{\widehat{\omega}_{ii}(\phi_j)\} = \omega_{ii} - \omega_{ii}^2$, the inequality to be satisfied simplifies to:

$$l \geq \log(\frac{2}{\delta})\left(\frac{2}{\epsilon^2\omega_{ii}} + \frac{2}{3\epsilon\omega_{ii}} - \frac{2}{\epsilon^2}\right).$$

Thus, selecting $l = \left\lceil \frac{6+2\epsilon}{3\sigma\epsilon^2} \ln\frac{2}{\delta} \right\rceil$ ensures the inequality always holds. This completes the proof. □

### A.5 Proof of Lemma 5.2

**Proof.** According to Lemma 4.2, $\boldsymbol{\rho}^i$ satisfying $\boldsymbol{\rho}^i = (I + D)^{-1}\boldsymbol{e}_i + (I + D)^{-1}A\boldsymbol{\rho}_i$. Thus, $\boldsymbol{e}_i^\top \boldsymbol{\rho}^i = \boldsymbol{e}_i^\top (I + D)^{-1}\boldsymbol{e}_i + \boldsymbol{e}_i^\top (I + D)^{-1}A\boldsymbol{\rho}^i$, that is $\omega_{ii} = \frac{1}{1+d_i}(1 + \sum_{j \in N(i)} \omega_{ji})$. Since $\widetilde{\omega}_{ii}(\phi) = \frac{1}{1+d_i}(1 + \sum_{j \in N(i)} \widehat{\omega}_{ji}(\phi))$, and $\widehat{\omega}_{ji}(\phi)$ is an unbiased estimator of $\omega_{ji}$, $\widetilde{\omega}_{ii}$ is an unbiased estimator of $\omega_{ii}$.

Next we aim to derive the variance of the estimator $\widetilde{\omega}_{ii}$. For a spanning converging forest $\phi \in \mathcal{F}$ and node $i, j \in V, i \neq j$, we have $\widehat{\omega}_{ij}(\phi) = \mathbb{I}_{\{r_\phi(i)=j\}} = 1$ or 0, leading to $\mathbb{E}\{\widehat{\omega}_{ij}^2\} = \mathbb{E}\{\widehat{\omega}_{ij}\} = \omega_{ij}$. Moreover, for distinct nodes $i, j, k \in V$, we have $\mathbb{E}\{\widehat{\omega}_{ji}(\phi)\widehat{\omega}_{ki}(\phi)\} = q_{jk}^{(i)}$. Thus, we can derive that

$$\text{Var}\{\widetilde{\omega}_{ii}(\phi)\} = \mathbb{E}\{(\widetilde{\omega}_{ii})^2\} - (\mathbb{E}\{\widetilde{\omega}_{ii}\})^2$$

$$= \frac{1}{(1+d_i)^2}\mathbb{E}\{(1 + \sum_{j \in N(i)} \widehat{\omega}_{ji})^2\} - \omega_{ii}^2$$

$$= \frac{1}{(1+d_i)^2}\mathbb{E}\{1 + 2\sum_{i \in N(j)} \widehat{\omega}_{ji} + (\sum_{i \in N(j)} \widehat{\omega}_{ji})^2\} - \omega_{ii}^2$$

$$= \frac{1 + 3\sum_{i \in N(j)} \omega_{ji}}{(1+d_i)^2} + \frac{2\sum_{j,k \in N(i)} q_{jk}^{(i)}}{(1+d_i)^2} - \omega_{ii}^2 \quad (8)$$

$$= \frac{1 + 3((1+d_i)\omega_{ii} - 1)}{(1+d_i)^2} + \frac{2\sum_{j,k \in N(i)} q_{jk}^{(i)}}{(1+d_i)^2} - \omega_{ii}^2$$

$$= \frac{3\omega_{ii}}{1+d_i} - \frac{2}{(1+d_i)^2} + \frac{2\sum_{j,k \in N(i)} q_{jk}^{(i)}}{(1+d_i)^2} - \omega_{ii}^2.$$

For a spanning converging forest $\phi \in \mathcal{F}$, and node $i, j \in V, i \neq j$, we consider the scenario where both $\widehat{\omega}_{ji}(\phi)$ and $\widehat{\omega}_{ki}(\phi)$ are equal to 1. This implies that node $i$ acts as a root node in $\phi$, leading to the conclusion $\widehat{\omega}_{ii} = 1$. From this observation, we can infer that the expected value of the product $\widehat{\omega}_{ji}(\phi)\widehat{\omega}_{ki}(\phi)$ is bounded by the expected value of $\widehat{\omega}_{ii}(\phi)$. Formally, this can be expressed as $q_{jk}^{(i)} \leq \omega_{ii}$. Building upon this foundation, we can further expand our analysis to compare the variances of the two estimators $\widehat{\omega}_{ii}$ and $\widetilde{\omega}_{ii}^{(0)}$. The mathematical derivation is as follows:

$$\text{Var}\{\widehat{\omega}_{ii}\} - \text{Var}\{\widetilde{\omega}_{ii}\} = \omega_{ii} - \frac{3\omega_{ii}}{1+d_i} + \frac{2}{(1+d_i)^2} - \frac{2\sum_{j,k \in N(i)} q_{jk}^{(i)}}{(1+d_i)^2}$$

$$\geq \omega_{ii} - \frac{3\omega_{ii}}{1+d_i} + \frac{2}{(1+d_i)^2} - \frac{d_i(d_i-1)\omega_{ii}}{(1+d_i)^2}$$

$$= \frac{2(1-\omega_{ii})}{(1+d_i)^2} \geq 0. \tag{9}$$

This derivation solidifies our understanding and confirms the reduced variance of the estimator $\widetilde{\omega}_{ii}$ compared to $\widehat{\omega}_{ii}$, thus completing our proof. $\square$

## A.6 Proof of Lemma 6.1

**Proof.** Since $\boldsymbol{\gamma}^i$ is the equilibrium vector of the iteration equation (3), $\boldsymbol{\gamma}^i$ satisfying $\boldsymbol{\gamma}^{i\top} = \boldsymbol{e}_i^\top (I+D)^{-1} + \boldsymbol{\gamma}^{i\top} A(I+D)^{-1}$. Thus, $\boldsymbol{\gamma}^{i\top}\boldsymbol{e}_i = \boldsymbol{e}_i^\top (I+D)^{-1}\boldsymbol{e}_i + \boldsymbol{\gamma}^{i\top} A(I+D)^{-1}\boldsymbol{e}_i$, that is $\omega_{ii} = \frac{1}{1+d_i}(1 + \sum_{j:i \in N(j)} \omega_{ij})$. Since $\overline{\omega}_{ii}(\phi) = \frac{1}{1+d_i} + \frac{1}{1+d_i}\sum_{j:i \in N(j)} \widehat{\omega}_{ij}(\phi)$., and $\widehat{\omega}_{ij}(\phi)$ is an unbiased estimator of $\omega_{ij}$, $\overline{\omega}_{ii}$ is an unbiased estimator of $\omega_{ii}$.

To determine the variance of the random variable $\overline{\omega}_{ii}(\phi)$, we can follow a similar approach as used for $\widetilde{\omega}_{ii}(\phi)$ in Lemma 5.2. It's crucial to note that for distinct nodes $i, j, k \in V$, the relationship $\widehat{\omega}_{ij}(\phi)\widehat{\omega}_{ik}(\phi) = 0$ consistently holds for all $\phi \in \mathcal{F}$. This ensures that the cross-product term is eliminated from our calculations. Given that the cross-product term is non-negative, it's evident that $\overline{\omega}(\phi)$ exhibits a reduced variance in comparison to the previous two random variables. $\square$

## A.7 Proof of Lemma 6.2

**Proof.** According to Lemma 6.1, we derive that

$$\left|\frac{\text{Var}\{\overline{\omega}_{ii}\}}{\omega_{ii}^2}\right| = \frac{3}{(1+d_i)\omega_{ii}} - \frac{2}{(1+d_i)^2\omega_{ii}^2} - 1$$

$$= -\frac{2}{(1+d_i)^2}\left(\frac{1}{\omega_{ii}} - \frac{3(1+d_i)}{4}\right)^2 + \frac{1}{8} \leq \frac{1}{8} \tag{10}$$

The equation holds when $\frac{1}{\omega_{ii}} = \frac{3(1+d_i)}{4}$, implying $\omega_{ii} = \frac{4}{3(1+d_i)}$, which finishes the proof. $\square$

## A.8 Proof of Theorem 6.3

**Proof.** For a given spanning converging forest $\phi \in \mathcal{F}$, and a node $i \in V$, the random variable $\overline{\omega}_{ii}(\phi)$ is defined as $\overline{\omega}_{ii}(\phi) = \frac{1}{1+d_i} + \frac{1}{1+d_i}\sum_{j:i \in N(j)} \widehat{\omega}_{ij}(\phi)$. Given that node $i$ has only one root node in $\phi$, the possible values of $\overline{\omega}_{ii}(\phi)$ are either $\frac{1}{1+d_i}$ or $\frac{2}{1+d_i}$. Since $\frac{1}{1+d_i} \leq \omega_{ii} \leq \frac{2}{2+d_i}$, it follows that $|\overline{\omega}_{ii} - \omega_{ii}| \leq \frac{1}{1+d_i}$.

In the context of Algorithm 3, we generate a set of $l$ spanning forests, denoted as $\phi_1, \cdots, \phi_l$. The output $\overline{\omega}[i]$ of Algorithm 3 is then expressed as: $\overline{\omega}[i] = \frac{1}{l}\sum_{j=1}^{l} \overline{\omega}_{ii}(\phi_j)$.

Now we can apply the Chernoff bound, as detailed in Lemma A.1. By setting $x_j = \overline{\omega}_{ii}(\phi_j)$ for $1 \leq j \leq l$, $M = \frac{1}{1+d_i}$, and $x = \overline{\omega}[i]$. Using the similar analysis as in the proof of Theorem 4.3, the number of $l$ needs to satisfy the following inequality:

$$l \geq \log\left(\frac{2}{\delta}\right)\left(\frac{2\text{Var}\{\overline{\omega}_{ii}\}}{\epsilon^2\omega_{ii}^2} + \frac{2}{3(1+d_i)\epsilon\omega_{ii}}\right).$$

Using Lemma 6.2 we have that $\frac{\text{Var}\{\overline{\omega}_{ii}\}}{\omega_{ii}^2} \leq \frac{1}{8}$ and $(1+d_i)\omega_{ii} \leq 1$. Thus, selecting $l = \left\lceil \left(\frac{2}{3\epsilon} + \frac{1}{4\epsilon^2}\right)\log\left(\frac{2}{\delta}\right)\right\rceil$ ensures the inequality always holds. This completes the proof. $\square$

