# OpenReview forum: "Efficient Computation for Diagonal of Forest Matrix via Variance-Reduced Forest Sampling"
_ACM.org/TheWebConf/2024/Conference — TheWebConf24 Oral_

### Official Review · Reviewer_Nn6r · 2023-11-22

**Novelty:** 6
**Technical Quality:** 6

**Review:**

**Summary**
This work proposes three new algorithms for estimating the diagonal entries of
the forest matrix for directed graphs: SCF, SCFV, and SCFV+. These samplers are
based on Wilson's algorithm and a probabilistic interpretation of the diagonal
entries (Theorem 4.1). Concretely, the forest matrix is
$\Omega = (I + L)^{-1}$ where $L$ is the graph Laplacian.
The inverse of the $i$-th diagonal entry of $\Omega$ is the average size of the
tree containing $i$ across all converging spanning forests where $i$ is a root
node.
All of these estimators are unbiased, and the relative variance of SCFV+ is
bounded by a constant (Lemma 6.2).

Previous works for estimating these diagonal entries are based on (undirected)
Laplacian solvers:
  * JLT (Jin et al., ICDM 2019)
  * UST (van der Grinten et al., ICDM 2021)
This work offers comprehensive experiments comparing their three algorithms
with JLT and UST (for undirected graphs), for a wide range of graph sizes.

Overall, this is a very strong paper, both theoretically and experimentally.
See the questions box for potential weaknesses and suggested improvements.

**Typos and suggestions**
- [line  043] suggestion: Start by defined $L$ in terms of the adjacency matrix
  of graph $G=(V,E)$ for unfamiliar readers.
- [line 124] typo: Missing space in "PageRank[22]"
- [line 171] suggestion: Replace "In the sequel," by "For brevity,"
- [line 183] suggestion: Italicize "rooted converging tree"
- [line 193] suggestion: Replace "surpass" by "dominate"
- [line 292] suggestion: Use $\mathbb{E}[\cdot]$ notation for expected values,
  i.e., brackets not braces.
- [line 350] typo: Strange behavior with ":" in the algorithm input/output. Same
  for Algorithm 2
- [line 354] suggestion: Just initialize with the vector value $\mathbf{0}_n$.
- [line 426] typo: $i$-th is boldfaced
- [line 792] typo: "average(upper)" --> "average (upper)"

**Questions:**

- [line 256] Is Theorem 4.1 novel? It seems that this equation should exist in
  the (directed) spectral graph theory literature. If so, would you cite the
  original reference?
- [line 333] Is the probability distribution on spanning converging forests as
  described in Section 4.2 really uniform? In particular, does Wilson's
  algorithm sample rooted trees converging trees from $G'$ uniformly? This could
  use a proof or citation from [4, 37,44].
- [line 390] Why are we writing the lower bound for $\omega_{ii}$ in terms of
  $\sigma$ when we also know that $\omega_{ii} \le \frac{1}{1 + d_i}$?
  Can this be tight? How do we estimate $\sigma$ when there's room for improvement?
- [line 530] If the variance of SCFV is lower than SCF (Lemma 5.2), why can't
  we use the bound for $l$ in SCF? Is the discussion here more about not being
  able to come up with a smaller value of $l$?
- [line 711] Does "average relative error" mean arithmetic or geometric mean?
- [line 847] How long do the "ground truth" computations take with GMRES?

**Reviewer Confidence:**

3: The reviewer is confident but not certain that the evaluation is correct

**Scope:**

4: The work is relevant to the Web and to the track, and is of broad interest to the community

---

### Official Review · Reviewer_CpSa · 2023-11-25

**Novelty:** 5
**Technical Quality:** 5

**Review:**

This paper is about estimating the diagonal entries of the matrix (I+L)^{-1} (also known as the forest matrix), where I is the identity matrix and L is the Laplacian matrix associated with an input graph. The problem itself is fundamental, lies at the core of network science, and has found several applications in Markov processes, opinion dynamics, graph signal processing, etc.

Compared to the state-of-the-art, where the emphasis has been on undirected graphs, the focus of this paper is on *directed* graphs. The core idea is to leverage the probabilistic interpretation of the diagonal elements of the forest matrix. The paper then presents a sampling estimation technique that builds upon and extends the seminal work of Wilson that uses loop-erased random walks for generating random spanning trees. A direct application of this yields accurate estimates only when the diagonal entries are large enough. To overcome this, the paper presents a new estimator with reduced variance and also argues about the quality of such an estimator.

The part of the paper concerning the experimental evaluations looks extensive and adequate. The proposed unbiased estimator appears to perform well and scales to relatively large networks.

*Pros
-The paper studies a very well-motivated and fundamental problem that seems to be important for many applications.
-Some effort has been put into trying to give provable guarantees for the presented algorithms, which is always welcome.

*Cons
-Unfortunately, I think the paper does make some inadequate claims, and some of the sentences/notations are a bit non-standard and hard to understand. All of these indicate that the paper may not be ready for wider dissemination. I have elaborated more in the comments to the authors below.
-Also, for the particular problem studied in the paper, I’m not very much convinced that directed graphs are harder than undirected ones.
-The approach presented here is also very similar to other works, e.g, [45]

-Evaluation
Overall, I think this is a sold contribution but due to my concerns above, I’m not that sure that it passes the bar for the WebConference.

**Questions:**

Generic comment: If you assume you have access to a very efficient directed Laplacian solver, what's then the main contribution of your work?

-In the abstract, you said “encounter limitations when applied to digraphs due to incapacity of the Laplacian solver”. What does this sentence mean? There are also directed Laplacian solvers presented in the literature (https://arxiv.org/abs/1811.10722) – why can’t you use those? Or what about practical solvers like multi-grid?
- Lines 52,53, the sentences are not connected well with each other.
- Line 88, what does “rooted probability” mean?
- Line 92, what does “sampling number” mean?
- Line 70, the Laplacian solver you cited is not the state-of-the-art Laplacian solver. Please do a more thorough literature review
- Line 137, again, Laplacian solvers for directed graphs exist
- Generic comment: why did you use spanning converging forests for describing forests of directed graphs? I think you might need to check the notion of pseudoforest online – it seems that’s what you need
- Line 215, a bad notation to use $\phi$ for a forest – it’s standard to use capital letters (and not Greek letters) for defining such objects
- Line 525, what’s a requisite sampling number?

**Reviewer Confidence:**

4: The reviewer is certain that the evaluation is correct and very familiar with the relevant literature

**Scope:**

4: The work is relevant to the Web and to the track, and is of broad interest to the community

---

### Official Review · Reviewer_FVNU · 2023-11-27

**Novelty:** 5
**Technical Quality:** 5

**Review:**

The paper studies the computation of the diagonal elements of the forest matrix $\Omega$ for directed graphs, where $\Omega = (I+L)^{-1}$ with $I$ being the identity matrix and $L$ being the graph Laplacian. These diagonal elements have implications of the structural importance of the corresponding nodes in the network. Previous nearly-linear algorithms are known for undirected graph based on the nearly linear time Laplacian solver (for undirected graphs). This paper gives three sampling based approximation algorithms SCF, SCFV and SCFV+ for directed graphs, and the best one SCFV+ can approximate all diagonal element within a multiplicative error of $\epsilon$ with probability at least $1-\delta$ in expected running time $O(\frac{n}{\epsilon^2}\log\frac{2}{\delta})$. This matches (or even slightly better) than the algorithm for undirected graphs, and is conceptually simpler as it doesn't involve Laplacian solver.

The basic idea is an interpretation of the diagonal element $\omega_{ii}$ as the probability that node $i$ is a root node in a randomly sampled spanning converging forest. The SCF algorithm faithfully follow this interpretation by randomly sampling $\ell$ spanning converging forests and using the empirical mean of node $i$ being a root as the estimated $\omega_{ii}$ for any $i$. To sample a random spanning converging forest, the algorithm adds dummy node $x$ and edges from every original node directed toward $x$, then uses the classic loop-erasing algorithm of Wilson to sample a directed spanning tree rooted at $x$, and finally remove $x$ to get the forest. Chernoff bound suggests when $\ell = \Omega(\frac{1}{\sigma\epsilon^2}\log\frac{2}{\delta})$ suffices when $\omega_{ii}\geq \sigma$ for all $i$. The further improve the sampling complexity and avoid the dependence on $\sigma$, the author(s) device two variance reduction techniques, and both are based on a known property of the $\Omega$ matrix from the opinion evolution literature. In particular, if one starts with some initial configuration $z(0)$ and follow a simple random walk type dynamic (parametrized by some seed vector $s$, then $z(0),z(1),\ldots,z(t)$ converges to $z=\Omega\cdot s$. This suggests that $\omega_i = \Omega\cdot e_i$, and if one can start with a $z(0)$ that is a fairly good estimation of $\omega_i$, then running a small number (even 1 step) of the dynamic can improve the quality of the estimate. The main idea of the improved algorithms SCFV and SCFV+ is to start with the estimation of SCF and run 1 step of the dynamic (or its transpose variant). For SCFV+ the authors can show a provable guarantee that $\ell=\Omega(\frac{1}{\epsilon^2}\log\frac{2}{\delta})$ is sufficient.

In the empirical evaluations, on undirected graphs the new algorithms show better approximation error (both on average and max) over previous algorithms under comparable complexity. On directed graphs, for several dataset of small size, the authors were able to compute the exact solution using numerical methods to demonstrate good approximation quality of their algorithm, and on large graphs (without ground truth to measure quality), the new algorithms (especially SCFV and SCFV+) are shown to scale reasonably well.

**Questions:**

I don't have any specific question.

**Reviewer Confidence:**

2: The reviewer is willing to defend the evaluation, but it is likely that the reviewer did not understand parts of the paper

**Scope:**

3: The work is somewhat relevant to the Web and to the track, and is of narrow interest to a sub-community

---

### Official Review · Reviewer_GvTg · 2023-11-28

**Novelty:** 5
**Technical Quality:** 5

**Review:**

This paper considers the problem of approximating the diagonal of the forest matrix. The forest matrix has many applications, such as in the computation of the forest closeness centrality. This matrix can be computed exactly by inverting the Laplacian, but this approach is expensive on large graphs. Previous methods for approximate computation of diagonal entries of the forest matrix leverage fast Laplacian solvers, which are only available for undirected graphs.

In this paper an alternative interpretation of the $i$-th diagonal entry is considered: its value is the fraction of converging spanning trees such that $i$ belongs is one of the roots. A natural idea is then to estimate this value as the fraction of randomly sampled spanning trees that have $i$ as one of the roots.

Based on this idea, the paper proposes three estimators of gradually smaller variances, and derives accuracy bounds via the Chernoff bound. The variance-reducing techniques are interesting.

In general, the description of the algorithms and estimators is clear.

On the other hand, the motivation for the importance of considering the diagonal entries of the forest matrix is implicitly assumed, with an extremely limited discussion. For example, in line 125: "The calculation of forest closeness centrality is inherently tied to the diagonal elements of the forest matrix"; how are they related? I would discuss more clearly how to use such entries to compute/approximate the forest closeness centrality, providing a clear motivation for the considered problem. Are there other applications where having accurate estimates of the diagonal is important? Could you provide more details on this?

The new algorithms are tested in practice on several large real-world graphs; the best-proposed method (SCFV+) is shown to be much more accurate than previous methods for the same amount of work (the sampling number $l$).

Regarding the experiments on directed graphs: the results are shown for fixed values of the number of samples $l$. When $l=1000$, the time to conclude for SCFV+ is approx 23 minutes. Are these values of $l$ useful to derive any approximation guarantee, as derived in the theoretical analysis (e.g., Theorem 6.3)?

Minor comments:
- line 69: memory instead of memories
- line 99: "superior theoretical result" is not very clear
- the used chernoff bound (Lemma A.1) seems to be Bernstein's inequality?

**Questions:**

- provide more details of at least one application where the diagonal of the forest matrix is important (e.g., forest closeness centrality)
- add more details on the obtained approximation guarantees for the considered parameters in the experiments

**Ethics Review Description:**

No issues

**Reviewer Confidence:**

3: The reviewer is confident but not certain that the evaluation is correct

**Scope:**

4: The work is relevant to the Web and to the track, and is of broad interest to the community

---

### Official Review · Reviewer_sTM8 · 2023-12-01

**Novelty:** 5
**Technical Quality:** 5

**Review:**

The paper studies the problem of estimating the diagonal entries of $(I+L)^{-1}$ where L is the Laplacian matrix of a directed graph. This matrix is called the forest matrix and it has some important applications that are mentioned in the paper. The authors give new variance-reduced techniques to estimate these quantities efficiently. The algorithms presented are necessary since the standard fast algorithms are constrained by fast directed Laplacian solvers.

Pros:
 Algorithm is simple and fairly easy to analyze. It has good practical performance and this seems to be important.

Cons:
It is not clear to me why are these methods required. There are near linear time laplacian solvers for directed graphs as well and the significance of this particular work is unclear.

**Questions:**

1. The comparison to previous work on directed graphs is unclear.
2. The matrix (I+L) should be diagonally dominant and I am wondering why isn't there an easier method to do this via some power iterations?
3. There are fast directed Laplacian solvers and it is unclear why the previous works do not work. Also, the barrier between the directed and undirected graphs is unclear. More precisely, why is the problem harder in directed graphs for this setting?

It seems to me that the main reason these new algorithms are required is due to the directed structure of the graph and not the Laplacian solvers. It would be good if the authors can clarify what exactly is the need for these.

Otherwise, the algorithms are presented fairly well and are easy to follow.

**Reviewer Confidence:**

3: The reviewer is confident but not certain that the evaluation is correct

**Scope:**

3: The work is somewhat relevant to the Web and to the track, and is of narrow interest to a sub-community

---

### Decision · Program_Chairs · 2024-01-22

**Decision:**

Accept (Oral)

**Comment:**

The reviewers agreed on the importance and broad applicability of the problem, that of obtaining the diagonal entries of the forest matrix of a directed graph. There was some discussion about whether sufficiently efficiently theoretical algorithms already exist for this problem, and that the current paper does not do a convincing job of highlighting the shortcomings of these algorithms from a practical perspective. However, the new algorithm proposed is simple and the rigorous analysis presented well. Moreover, the experiments convincingly establish the practical utility of the algorithm vis-a-vis previous theoretical work on undirected graph, i.e., in a simpler setting than directed graphs. Overall, the reviewers were positive about the paper, although not very enthusiastic. The paper can be (weakly) recommended for acceptance.